# A Retinal Oct-Angiography and Cardiovascular STAtus (RASTA) Dataset of Swept-Source Microvascular Imaging for Cardiovascular Risk Assessment

Clément Germanèse [1,2], Fabrice Meriaudeau [2], Pétra Eid [1], Ramin Tadayoni [3], Dominique Ginhac [2], Atif Anwer [2], Steinberg Laure-Anne [1], Charles Guenancia [4,5], Catherine Creuzot-Garcher [1], Pierre-Henry Gabrielle [1] and Louis Arnould [1,5,*]

1   Department of Ophthalmology, Dijon University Hospital, 21079 Dijon CEDEX, France; clement.germanese@chu-dijon.fr (C.G.); petra.eid@chu-dijon.fr (P.E.); laure-anne.steinberg@chu-dijon.fr (S.L.-A.); catherine.creuzot-garcher@chu-dijon.fr (C.C.-G.); pierrehenry.gabrielle@chu-dijon.fr (P.-H.G.)
2   Artificial Vision and Imaging (ImViA), Imagerie Fonctionnelle et Moléculaire et Traitement des Images Médicales (IFTIM), (EA 7535), Faculty of Health Sciences, Université de Bourgogne Franche-Comté, 21078 Dijon, France; fabrice.meriaudeau@u-bourgogne.fr (F.M.); dominique.ginhac@ubfc.fr (D.G.); atif.anwer@u-bourgogne.fr (A.A.)
3   Department of Ophthalmology, Université Paris Cité, AP-HP, Lariboisière, Saint Louis and Adolphe de Rothschild Fondation Hospitals, 75000 Paris, France; ramin.tadayoni@aphp.fr
4   Department of Cardiology, Dijon University Hospital, 21079 Dijon CEDEX, France; charles.guenancia@chu-dijon.fr
5   Pathophysiology and Epidemiology of Cerebro-Cardiovascular Diseases (PEC2), (EA 7460), Faculty of Health Sciences, Université de Bourgogne Franche-Comté, 21000 Dijon, France
*   Correspondence: louis.arnould@chu-dijon.fr

**Abstract:** In the context of exponential demographic growth, the imbalance between human resources and public health problems impels us to envision other solutions to the difficulties faced in the diagnosis, prevention, and large-scale management of the most common diseases. Cardiovascular diseases represent the leading cause of morbidity and mortality worldwide. A large-scale screening program would make it possible to promptly identify patients with high cardiovascular risk in order to manage them adequately. Optical coherence tomography angiography (OCT-A), as a window into the state of the cardiovascular system, is a rapid, reliable, and reproducible imaging examination that enables the prompt identification of at-risk patients through the use of automated classification models. One challenge that limits the development of computer-aided diagnostic programs is the small number of open-source OCT-A acquisitions available. To facilitate the development of such models, we have assembled a set of images of the retinal microvascular system from 499 patients. It consists of 814 angiocubes as well as 2005 en face images. Angiocubes were captured with a swept-source OCT-A device of patients with varying overall cardiovascular risk. To the best of our knowledge, our dataset, Retinal oct-Angiography and cardiovascular STAtus (RASTA), is the only publicly available dataset comprising such a variety of images from healthy and at-risk patients. This dataset will enable the development of generalizable models for screening cardiovascular diseases from OCT-A retinal images.

**Dataset:** https://rasta.u-bourgogne.fr/

**Dataset License:** CC-BY 4.0

**Keywords:** retina; swept-source; optical coherence tomography angiography; cardiovascular risk; $CHA_2DS_2$-VASc

## 1. Summary

Cardiovascular diseases (CVD) remain the leading cause of death worldwide with nine million deaths from heart disease reported in 2019 [1]. Pathophysiological mechanisms

involved in the development of CVD begin years before the appearance of any symptoms [2]. Thus, researchers have been investigating early biomarkers to help screen and diagnose CVD before the onset of symptoms or major cardiovascular events. The retinal vascular network could be a good candidate since the retinal microvasculature may share the same physiological and anatomical characteristics as the cerebral and coronary microvasculature [3]. Associations between retina vascular features and CVD were first demonstrated with fundus photographs [4,5]. These associations were subsequently confirmed with other retinal imaging such as retinal swept source optical coherence tomography angiography (SS OCT-A) [6,7]. SS OCT-A enables the noninvasive assessment of the retinal microvascular network. It is thus possible to study the different vascular plexi (superficial capillary plexus, deep capillary plexus, and choriocapillaris plexus) and the avascular zone using quantitative data. The quantification of retinal vascular density by SS OCT-A could therefore be compared to a window into the integrity of the systemic microcirculation.

The cardiovascular risk profile of patients can be estimated with numerous score models such as the Framingham Risk score (FRS) for 10-year CVD risk calculation, the Pooled Cohort Equations (PCE), the American Heart Association risk score (AHA risk score) for a moderate-risk population, or the SCORE2 to predict the 10-year risk of first-onset CVD in European populations [8–11]. The $CHA_2DS_2$-VASc clinical score, which is universally known and easy to calculate, is an embolic risk stratification tool originally used to assess the risk of stroke in patients with non-valvular atrial fibrillation [12]. It has been recently presented as an effective model for evaluating the cardiovascular risk profile regardless of the arrhythmic status of patients [13–19]. Several datasets containing images of retinal fundus photographs are publicly available (i.e., MESSIDOR, STARE project, DRIVE, E-ophtha, and EyePACS) [20–24]. However, SS OCT-A datasets are less widespread [25]. To the best of our knowledge, the Retinal oct-Angiography and cardiovascular STAtus (RASTA) dataset is the first publicly available dataset that provides systematic cardiovascular data and complete SS OCT-A retinal imaging. The RASTA dataset is hosted on https://rasta.u-bourgogne.fr/, accessed on 28 September 2023.

## 2. Ethics Approval

The RASTA dataset was acquired from the Department of Ophthalmology at the University Hospital of Dijon, France, and consists of actual clinical acquisitions from different registered clinical studies. The RASTA dataset was anonymized and processed in accordance with the rules established by the Ethics Committee of the University Hospital of Dijon. All administrative information included in the metadata has been removed, making it untraceable. Thus, in accordance with the French law it was not necessary to obtain ethical approval.

## 3. Data Description

### 3.1. Data Composition

The RASTA dataset is a new publicly available SS OCT-A retinal image dataset consisting of 499 participants for 2005 en face images and 814 angiocubes combined with clinical and demographic characteristics. Information on data accessibility and specifications is provided in Table 1. Each participant was identified by an anonymized ID and was then included in one of three groups according to their cardiovascular risk category as follows:

- Low cardiovascular risk—$CHA_2DS_2$-VASc = [0; 1];
- Intermediate cardiovascular risk—$CHA_2DS_2$-VASc = [2; 3];
- High cardiovascular risk—$CHA_2DS_2$-VASc = [3; 9].

For each participant, we included the images of their corresponding SS OCT-A $6 \times 6$-mm angiocubes and en face (two-dimensional) images. Angiocubes were identified on the basis of their side only. En face images (Figure 1) were identified on the basis of their plexus followed by their side as follows:

- «sup» for superficial plexus or «deep» for deep plexus or «cc» for choriocapillaris plexus
- «OD» for right eye or «OS» for left eye

**Table 1.** Specifications table.

| | |
|---|---|
| Subject Area | Biomedical Imaging, Ophthalmology |
| More specific subject area | Retinal OCT-A volumes analysis for cardiovascular risk prediction |
| Type of data | Image, CSV |
| How data were acquired | Swept-source OCT-A<br>Instrument name: PLEX Elite 9000® (Carl Zeiss Meditec Inc., Dublin, OH, USA) |
| Data format | DICOM for volumes, Bitmap for en face images |
| Experimental factors | Pupillary dilatation with tropicamide 0.5% if signal strength < 8/10 |
| Experimental features | Macular angiography 6 × 6-mm |
| Main data source location | University Hospital of Dijon, Dijon 21000, France |
| Data accessibility | https://rasta.u-bourgogne.fr/, accessed on 28 September 2023. |

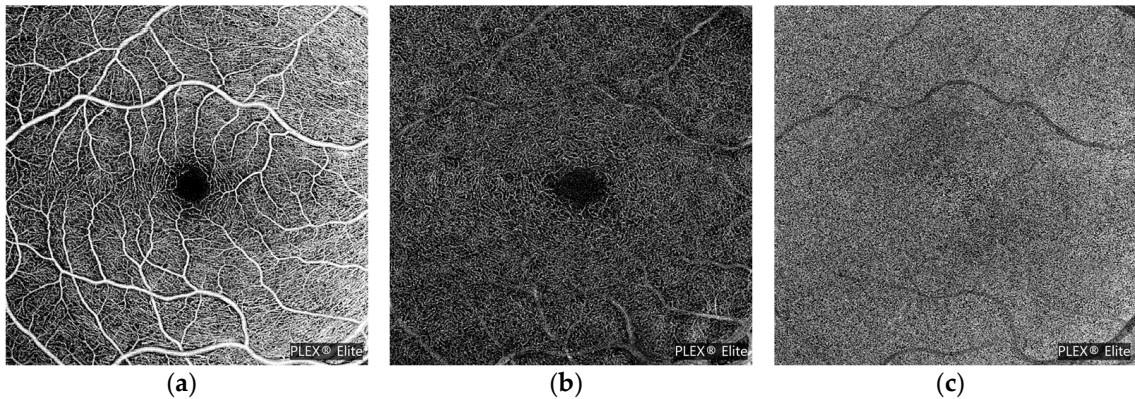

(**a**)        (**b**)        (**c**)

**Figure 1.** Right eye en face images of (**a**) superficial plexus, (**b**) deep plexus, and (**c**) choriocapillaris plexus.

The RASTA dataset is composed of four different single-center studies with the Ophthalmology Department of the University Hospital of Dijon as the principal investigator since 2018 and one multicenter study conducted by 14 investigative health centers since 2021. All of the studies required the collection of cardiovascular history and anthropometric data. The aims of these studies are described as follows:

1. «AnomAlies Rétiniennes précoces au cours du Diabète de type 1» (**AwARD**; Early Retinal Anomalies in Type 1 Diabetes) [26]: to specify early retinal microvascular abnormalities by measuring the area of the central retinal avascular zone on SS OCT-A images of patients with type 1 diabetes without diabetic retinopathy (ID-RCB: 2017-A02724-49); 95 eyes of 95 patients, from 02/23/2018 to 02/28/2020.

2. **RETINORM**: control group of the AwARD study; 137 eyes of 75 volunteers, from 04/12/2021 to 11/25/2021.

3. «Retinal Microvascular Changes in Familial Hypercholesterolemia: Analysis with Swept-Source Optical Coherence Tomography Angiography» (**FAMILIPO**) [27]: to analyze the association between retinal vascular density and the presence of atherosclerosis assessed with the Coronary Artery Calcium score and compare SS OCT-A quantitative parameters between patients with familial hypercholesterolemia (FH) and healthy volunteers from the AwARD study without a history of FH; 162 eyes of 81 patients with FH, from 10/21/2020 to 10/27/2021.

4. «Obstructive sleep apnea and Retinal vascular NETwork» (**ORNET**): to describe retinal microvascular characteristics with SS OCT-A in a population with obstructive sleep apnea syndrome (OSAS) and to compare these patients with healthy volun-

teers (ID-RCB: 2018-A02204-51); 159 eyes of 79 patients with OSAS and 62 eyes of 33 volunteers without OSAS, from 07/01/2020 to 02/14/2023.

5. «Réseau Microvasculaire Rétinien et Chirurgie Cardiaque de revascularisation coronarienne» (**MRCC**; Retinal Microvascular Network and Coronary Revascularization Cardiac Surgery): to study, in patients scheduled for coronary revascularization cardiac surgery with extracorporeal circulation, the discriminative capacity of the retinal vascular density to predict the occurrence of acute renal failure defined by the KDIGO criterion [28] within 7 days of surgery (ID-RCB: 2021-A02895-36); 33 eyes of 33 patients, from 06/07/2022 to 03/06/2023.

6. «Giant cell arteritis study» (**GIANT**): to describe retinal microvasculature on SS OCT-A in patients with giant cell arteritis without ophthalmological symptoms; 56 eyes of 40 patients, from 11/21/2017 to 10/18/2022.

7. «Evaluation intelligente de la Rétinopathie diabétique» (**EviRed**; Intelligent Assessment of Diabetic Retinopathy): to propose SS OCT-A analysis to better predict the risk of diabetic retinopathy than the current classification of diabetic retinopathy mainly based on fundus photography (ANR: 18-RHUS-0008); 118 eyes of 63 patients without diabetic retinopathy, from 06/01/2021 to 01/19/2022.

A CSV file contains each ID in alphanumerical order with the corresponding characteristics. Each medical diagnosis has been confirmed by a panel of medical experts according to the guidelines of the French National Authority for Health (Haute Autorité de Santé). The information available in the CSV file is illustrated in Table 2, with the following explanation for each column:

**Table 2.** Sample CSV files.

| ID | Age | Sex | Congestive Heart Failure | Hypertension | Diabetes Mellitus | Stroke | Vascular Disease | Body Mass Index | CHA$_2$DS$_2$-VASc | Obstructive Sleep Apnea Syndrome |
|---|---|---|---|---|---|---|---|---|---|---|
| 7BLCE82 | 39.3 | 1 | 0 | 0 | 1 | 0 | 0 | 27.63 | 2 | 0 |
| 7BODO57 | 63.7 | 0 | 0 | 1 | 1 | 0 | 0 | 39.71 | 2 | 0 |

| Smoking | Dyslipidemia | FAZ_RL_OD | FAZ_Ci_OD | FAZ_RS_OD | FAZ_RL_OS | FAZ_Ci_OS | FAZ_RS_OS | Dens_Ave_Sup_OD |
|---|---|---|---|---|---|---|---|---|
| 0 | 0 | 1.847828 | 0.7151644 | 0.1943207 | 1.870777 | 0.7684844 | 0.2140274 | 17.9124348958329 |
| 0 | 1 | 3.164148 | 0.5358088 | 0.4268875 | 1.629538 | 0.7863739 | 0.1661682 | 17.6601562499993 |

| Dens_Circle3mm_Sup_OD | Dens_Circle6mm_Sup_OD | Dens_Ave_Sup_OS | Dens_Circle3mm_Sup_OS | Dens_Circle6mm_Sup_OS |
|---|---|---|---|---|
| 15.7255366682872 | 17.2295695743654 | 20.0175781250004 | 19.2683353754627 | 19.9760718897393 |
| 14.4667042195168 | 17.5332242119224 | 18.0494791666661 | 18.1161517910436 | 17.8680600309991 |

| Perf_Ave_Sup_OD | Perf_Circle3mm_Sup_OD | Perf_Circle6mm_Sup_OD | Perf_Ave_Sup_OS | Perf_Circle3mm_Sup_OS |
|---|---|---|---|---|
| 0.398761749267578 | 0.335599805730937 | 0.380546984640812 | 0.436973571777344 | 0.399284283408211 |
| 0.380107879638672 | 0.304275197638055 | 0.372604011433318 | 0.409038543701172 | 0.400241226363707 |

| Perf_Circle6mm_Sup_OS | Dens_Ave_Deep_OD | Dens_Circle3mm_Deep_OD | Dens_Circle6mm_Deep_OD | Dens_Ave_Deep_OS |
|---|---|---|---|---|
| 0.431644794054866 | 8.82747395833345 | 5.98949651934599 | 8.44266017287466 | 15.5621744791656 |
| 0.398799313893654 | 6.07356770833347 | 4.00121723028324 | 5.5699215791659 | 4.33268229166667 |

| Dens_Circle3mm_Deep_OS | Dens_Circle6mm_Deep_OS | Perf_Ave_Deep_OD | Perf_Circle3mm_Deep_OD | Perf_Circle6mm_Deep_OD |
|---|---|---|---|---|
| 12.0008038845776 | 15.7933960523881 | 0.174694061279297 | 0.118523555123847 | 0.166902197033784 |
| 5.46489248566923 | 3.8770957475287 | 0.118579864501953 | 0.0750344774003069 | 0.106823345466983 |

| | | Perf_Ave_Deep_OS | Perf_Circle3mm_Deep_OS | Perf_Circle6mm_Deep_OS |
|---|---|---|---|---|
| | | 0.310855865478516 | 0.237508995079448 | 0.315187959522492 |
| | | 0.0877456665039063 | 0.105439265426815 | 0.07639377745169 |

- **ID**: participant's anonymous identity code.
- **Age**: age in years at inclusion.
- **Sex**: 0 if male gender, 1 if female gender.
- **Congestive heart failure**: presence of heart failure/moderate–severe cardiac dysfunction with left ventricular ejection fraction ≤ 40%.
- **Hypertension**: presence of hypertension confirmed by ambulatory blood pressure measurement with a systolic blood pressure ≥ 135 mmHg and/or diastolic blood pressure ≥ 85 mmHg.

- **Diabetes mellitus**: presence of diabetes mellitus confirmed by a single blood glucose sample $\geq$ 2 g/L or confirmed by a second blood glucose sample $\geq$ 1.26 g/L when the first one was $\geq$1.26 g/L and <2 g/L.
- **Stroke**: prior stroke or transient ischemic attack or thromboembolism.
- **Vascular disease**: presence of vascular disease (e.g., peripheral artery disease, myocardial infarction, aortic plaque) confirmed by Doppler ultrasonography, coronary angiography/cardiac magnetic resonance imaging (MRI)/myocardial perfusion scintigraphy, or computed tomography angiography.
- **Body mass index**: body mass divided by the square of height, in kg/m$^2$.
- **CHA$_2$DS$_2$-VASc**: cardiovascular score prediction.
- **Obstructive sleep apnea syndrome**: presence of obstructive sleep apnea syndrome confirmed by respiratory polygraphy or polysomnography.
- **Smoking**: previous or active smoking.
- **Dyslipidemia**: presence of dyslipidemia confirmed by two blood samples with HDL-c < 0.35 g/L or LDL-c > 1.30 g/L and/or TG > 1.5 g/L for patients with cardiovascular risk and two blood samples with HDL-c < 0.35 g/L or LDL-c > 1.60 g/L and/or TG > 1.5 g/L for patients without cardiovascular risk.
- **OD**: oculus dexter (right eye).
- **OS**: oculus sinister (left eye).
- Fovea Avascular Zone (**FAZ**) in superficial plexus:

  ○ FAZ_RL: raw length (perimeter) of the FAZ in mm;
  ○ FAZ_Ci: circularity index of the FAZ ranging from 0 (most irregular circular shape) to 1 (perfect circular shape);
  ○ FAZ_RS: raw size (area) of the FAZ in mm$^2$.

- Vessel **dens**ity (VD): total length of perfused vasculature per unit area in a region of measurement in units of mm$^{-1}$. It consists of untangling the entire vasculature in the retina, measuring its length, and then dividing it by the area it originally occupied, ranging from a minimum of 0 (no vessels) to an unbounded maximum.

  ○ Dens_Ave_Sup: VD average in the superficial plexus;
  ○ Dens_Circle3mm_Sup: VD in a circle of 3 mm diameter in the superficial plexus;
  ○ Dens_Circle6mm_Sup: VD in a circle of 6 mm diameter in the superficial plexus;
  ○ Dens_Ave_Deep: VD average in the deep plexus;
  ○ Dens_Circle3mm_Deep: VD in a circle of 3 mm diameter in the deep plexus;
  ○ Dens_Circle6mm_Deep: VD in a circle of 6 mm diameter in the deep plexus.

- **Perf**usion density (PD): total area of perfused vasculature per unit area in a region of measurement ranging from 0 (no perfusion) to 1 (fully perfused).

  ○ Perf_Ave_Sup: PD average in the superficial plexus;
  ○ Perf_Circle3mm_Sup: PD in a circle of 3 mm diameter in the superficial plexus;
  ○ Perf_Circle6mm_Sup: PD in a circle of 6 mm diameter in the superficial plexus;
  ○ Perf_Ave_Deep: PD average in the deep plexus;
  ○ Perf_Circle3mm_Deep: PD in a circle of 3 mm diameter in the deep plexus;
  ○ Perf_Circle6mm_Deep: PD in a circle of 6 mm diameter in the deep plexus.

### 3.2. Swept-Source OCT-A Acquisitions

OCT-A is a noninvasive imaging technique that provides three-dimensional visualization of the perfused vasculature of the retina and choroid. In contrast to standard structural OCT, OCT-A analyzes not only the intensity of the reflected light but also the temporal changes in the reflection caused by moving particles, such as erythrocytes flowing through vessels. These changes in the OCT signal are detected by repeatedly capturing OCT images at each point on the retina and allow for the creation of image contrast between perfused vessels and static surrounding tissues [29]. To acquire such data, various algorithms have been established by several manufacturers, making resultant images different in appearance from one another. Such variances in the output of each device may result in different

interpretations of the clinical diagnosis. More specifically, the success of an algorithm may be dependent on the number of repeated OCT scans at each retinal location and on the sensitivity of the algorithm to differentiate particles in motion from static tissue. In addition to these considerations, each device may also differ with regard to acquisition speed and the retinal boundaries that are applied to differentiate various vascular plexi (using en face images generated from slabs). Moreover, while each unique OCT-A algorithm is subject to slightly different limitations that are attributed to its overall approach, there are certain confounding factors and/or limitations that impact all algorithms and are innate characteristics of this imaging modality.

The acquisition of OCT-A volume scans provides a three-dimensional cube of data that includes structural OCT and OCT-A images. A series of OCT section images (or B-scans) are acquired in order to create this cube of data. An initial review of these data is usually based on images that are generated from slabs of the cube. Slabs are sections of three-dimensional volumetric data. In the case of OCT-A slabs, the section is delimited by anterior and posterior retinal and choroidal boundaries. The OCT-A signal between these boundaries is displayed as a two-dimensional en face image, showing perfused vasculature. It is referred to as an en face image due to the transversal slab orientation; the resulting image gives the impression of looking onto the retina.

With the PLEX Elite 9000® instrument from Zeiss (Carl Zeiss Meditec Inc., Dublin, OH, USA), a 6 × 6 mm (~21° × 21°) scan pattern provides a relatively large overview of the retinal and choroidal circulation, ideal for the detection of vascular abnormalities that may not be present in the avascular central macula. This high-speed scan has an isotropic lateral resolution of 11.7 μm/pixel (512 A-scans × 512 B-scans), and it can offer the resolution needed to visualize small capillaries. Considering the small diameter of these smallest capillaries (approximately 12 μm), a lower-resolution scan may limit the confidence or reliability in image interpretation. The specifications of the PLEX Elite 9000® instrument are shown in Table 3. Finally, this high-resolution scan facilitates a more detailed and confident evaluation of vascular abnormalities at the capillary level.

**Table 3.** OCT device specifications.

| Model | Manufacturer | Technology | Hardware |
|---|---|---|---|
| PLEX Elite 9000® | Carl Zeiss Meditec Inc, Dublin, OH, USA | Swept Source Optical Coherence Tomography | Optical Micro AngioGraphy (OMAG) |

| FOV | Wave Length | Slew Rate | Axial Scan Depth | Optical Axial Resolution | Optical Transversal Resolution | Number of Images in Dataset |
|---|---|---|---|---|---|---|
| 56° | 1040–1060 nm | 100,000 A-scans/sec | 3.0 mm | 6.3 μm | 20 μm | 2005 en face images 814 angiocubes |

### 3.3. Quantitative OCT-A Vascular Features

All angiocubes were segmented and analyzed on a cloud platform called the Advanced Research and Innovation Network (ARI Network). Quantification analysis was performed using the «Macular Vasculature Density v0.7.3.3» algorithm. This algorithm quantifies the vascular density (vessel and perfusion) of superficial and deep retina layers; it also quantifies the foveal avascular zone (FAZ) of the superficial layer. The outputs offered are:

- Superficial and deep slabs (angio and structure);
- Vessel and perfusion traces for superficial and deep slabs;
- Superficial and deep vessel and perfusion density maps, color overlay images;
- FAZ superficial segmentation;
- Density and FAZ quantification results.

*3.4. Cardiovascular Data*

Historical models for the prediction of CVD in the general population, such as FRS, PCE, and the recently updated AtheroSclerotic CVD (ASCVD) Risk Estimator Plus, may have some limitations when used for patients with an intermediate risk profile. The latest guidelines from the American College of Cardiology and American Heart Association (AHA) recommend the use of the ASCVD Risk Estimator Plus, which provides a 10-year CVD risk score based on certain risk factors (age, sex, ethnicity), bedside tests (e.g., blood pressure), and blood parameters (e.g., total cholesterol) [30]. However, even such risk stratification algorithms can have limited calibration and discriminative ability when externally validated [31,32]. Moreover, generating these scores requires invasive biological sampling and depends on significant input from healthcare professionals and laboratory testing.

The universally known $CHA_2DS_2$-VASc clinical score, which is simple and quick to calculate, is a risk stratification tool initially used to estimate the risk of stroke in people with non-rheumatic atrial fibrillation [12]. It is a risk factor-based approach that defines definitive risk factors (previous stroke/transient ischemic attack [TIA]/thromboembolism [TE] and age $\geq$ 75 years) and combination risk factors (heart failure/moderate–severe cardiac dysfunction, hypertension, diabetes, vascular disease, female gender, and age 65–74 years), as shown in Table 4. As we wished to artificially categorize the neurocardiovascular risk of these individuals, high risk was defined as the presence of one definitive or two or more combination risk factors, intermediate risk was essentially defined as the presence of one combination risk factor, and low risk was defined as the presence of one or no risk factor (Table 5). Guidelines from the AHA and the European Society of Cardiology recommend the use of this stratification system for the indication of oral anticoagulant therapy. However, because all components of the $CHA_2DS_2$-VASc score are important cardiovascular risk factors, a recent cohort study demonstrated that an incrementally higher $CHA_2DS_2$-VASc score is associated with stroke in patients regardless of the presence of atrial fibrillation and can help identify patients at higher risk of mortality [13–19,33–37]. To date, there is no consensus regarding the use of the $CHA_2DS_2$-VASc score for global cardiovascular risk stratification, but it appears that a high score of >3 would be synonymous with a high cardiovascular risk.

**Table 4.** $CHA_2DS_2$-VASc point-based scoring system.

| Risk Factor | Score |
|---|---|
| **C**ongestive heart failure/Left ventricular dysfunction | 1 |
| **H**ypertension | 1 |
| **A**ge $\geq$ 75 years | 2 |
| **D**iabetes mellitus | 1 |
| **S**troke/TIA/TE | 2 |
| **V**ascular disease (prior myocardial infarction, peripheral artery disease, or aortic plaque) | 1 |
| **A**ge 65–74 years | 1 |
| **S**ex **c**ategory (i.e., female gender) | 1 |

**Table 5.** Risk Scheme used for neurocardiovascular risk stratification.

| Risk Scheme | Low Risk [0; 1] | Intermediate Risk [2; 3] | High Risk [4; 9] |
|---|---|---|---|
| RASTA (2023) | One or no combination risk factor | One definitive risk factor and 1 or no combination risk factor, or 2 or 3 combination risk factors | Two definitive risk factors, or 1 definitive risk factor and $\geq$2 combination risk factors, or $\geq$4 combination risk factors |

Definitive risk factors: previous stroke/TIA/TE, age > 75; Combination risk factors: heart failure/left ventricular ejection fraction $\leq$ 40%, hypertension, diabetes, vascular disease, female gender, age 65–74.

## 4. Methods

Clinic and demographic data were collected at the inclusion of each participant in the study using a single medical interview common to each of the studies mentioned above. All information was verified by an investigating operator from the patients' hospital medical chart, if available.

Each participant underwent an SS OCT-A examination of one or both eyes using the PLEX Elite 9000®. Examinations were acquired by three different trained operators and were performed under standard dark conditions. Pupillary dilatation was systematically performed with one eye drop of tropicamide 0.5% in both eyes if the B-scan signal strength was lower than 8/10. A $6 \times 6$-mm PLEX Elite 9000® angiography examination was performed for each of the included eyes. Only acquisitions with a signal strength greater than or equal to 8/10 were processed. Each angiocube and en face image were reviewed by an ophthalmologist without knowledge of the participant's cardiovascular status. For volumetric acquisitions, if an acquisition was judged to be of poor quality or with too much noise after review, the participant was excluded from the database. En face images judged to be of insufficient quality by the ophthalmologist were deleted from the database.

## 5. Conclusions

Emerging modern imaging techniques such as SS OCT-A have created an unprecedented opportunity to comprehensively characterize the microscopic ophthalmic features associated with CVD, also known as oculomics [38,39]. This oculomics revolution has opened up new avenues, including the use of the retina to obtain insights beyond the eye. Detecting microvascular changes before clinical manifestations can have predictive value, and ophthalmoscopic changes in the retinal microvasculature structure with SS OCT-A might represent a unique opportunity to fulfill this task.

Here, we introduce the first existing dataset of SS OCT-A images combined with cardiovascular data. Our dataset called RASTA contains volumetric acquisitions from 499 patients and 2005 segmented en face images with corresponding quantitative microvascular features and clinical cardiovascular data. The main interest of the RASTA dataset lies in the hybrid nature of the data that can strengthen collaborative research between ophthalmology and cardiology and refine the correlation between the vascular retinal network and cardiovascular diseases. Open access to medical imaging datasets remains a huge challenge for the community, which hinders the development of deep learning-based solutions as they require large datasets to reach efficient performances.

**Author Contributions:** Conceptualization, F.M., L.A., C.G. (Clément Germanèse) and C.C.-G.; methodology, F.M.; software, D.G.; validation, L.A., F.M., P.-H.G., R.T. and C.C.-G.; investigation, C.G. (Clément Germanèse); resources, C.G.; data curation, C.G. (Clément Germanèse), P.E., A.A. and S.L.-A.; writing—original draft preparation, C.G. (Clément Germanèse); writing—review and editing, C.G. (Charles Guenancia), L.A. and F.M.; supervision, L.A. and F.M.; project administration, F.M. and C.C.-G. All authors have read and agreed to the published version of the manuscript.

**Funding:** This research received no external funding.

**Institutional Review Board Statement:** The RASTA dataset was obtained in accordance with the Declaration of Helsinki. The RASTA dataset was anonymized and processed in accordance with the rules established by the Ethics Committee of the University Hospital of Dijon. All administrative information included in the metadata has been removed, making it untraceable. Thus, in accordance with the French law it was not necessary to obtain ethical approval.

**Informed Consent Statement:** Informed consent was obtained from all subjects involved in the study.

**Data Availability Statement:** https://rasta.u-bourgogne.fr/, accessed on 28 September 2023.

**Conflicts of Interest:** The authors indicate no financial support specifically for this study. C. Germanese has nothing to disclose. F. Meriaudeau has nothing to disclose. P. Eid has nothing to disclose. R. Tadayoni has received grants from Novartis, Abbvie, Bayer, and Alcon, personal fees from Novartis, Abbvie, Roche, Bayer, Alcon, Théa, Apellis, Iveric Bio, and Oculis and non-financial support

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
