# Peer review of "A Retinal Oct-Angiography and Cardiovascular STAtus (RASTA) Dataset of Swept-Source Microvascular Imaging for Cardiovascular Risk Assessment"

_data, 2023_

Round 1

Reviewer 1 Report (Previous Reviewer 3)

Give the detailed data and methods of the whole study group.

Confused, mixed data of patients from several studies not comparable

poor

Author Response

Reviewer 1
Comments and Suggestions for Authors
Give the detailed data and methods of the whole study group.
Confused, mixed data of patients from several studies not comparable

We thank the Reviewer for these comments. The detailed data and methods have been
presented in the methods section with presentation of each dataset (eg. Retinorm, MRCC).
Regarding the mixed data comment this is precisely the point of a dataset repository

Reviewer 2 Report (Previous Reviewer 2)

The authors’ work seems very meaningful. However, the confusing structure makes it difficult to understand the author's work. Here below are some specific questions or suggestions:

1.       Generally, the first section is an introduction. And at present, this section is too limited to show the development of this area. And the structure of the paper should be clear at the end of the introduction.

2.       The logic of the structure should be: Introduction, Methods, Data Description, Results and Discussion (including figures and tables), Conclusions.

3.       List of acronyms must be provided before introductions.

4.       Please clearly highlight how your work advances the field from the present state of knowledge and you should provide a clear justification for your work. The impact or advancement of the work can also appear in the conclusion.

Author Response

Reviewer 2
Comments and Suggestions for Authors
The authors’ work seems very meaningful. However, the confusing structure makes it
difficult to understand the author's work. Here below are some specific questions or
suggestions:
Generally, the first section is an introduction. And at present, this section is too limited to
show the development of this area. And the structure of the paper should be clear at the
end of the introduction.

We thank the Reviewer for his comments. We are not presenting an original research
paper. It is an opensource dataset repository. The aim of this project is to allow
collaborative work on this specific field of research. This is the first complete dataset with
retinal imaging and good quality labeled cardiovascular data. Please refer to the
submission guidelines of this specific type of manuscript in data journal.

The logic of the structure should be: Introduction, Methods, Data Description, Results and Discussion (including figures and tables), Conclusions.

Please refer to the submission guidelines of this specific type of manuscript in data journal.

List of acronyms must be provided before introductions.

Please refer to the submission guidelines of this specific type of manuscript in data journal.

Please clearly highlight how your work advances the field from the present state of knowledge and you should provide a clear justification for your work. The impact or advancement of the work can also appear in the conclusion.

We have already published several datasets in this journal. This dataset could motivate
collaboration in this very dynamic field of research. Opensource dataset are mandatory for good quality research in Artificial Intelligence.

Retinal Fundus Multi-Disease Image Dataset (RFMiD): A Dataset for Multi-Disease Detection Research by Samiksha Pachade,Prasanna Porwal,Dhanshree Thulkar,Manesh Kokare,Girish Deshmukh,Vivek Sahasrabuddhe,Luca Giancardo,Gwenolé Quellec andFabrice Mériaudeau
Data 2021, 6(2), 14; https://doi.org/10.3390/data6020014 - 03 Feb 2021
Cited by 50 | Viewed by 14695

A Tumour and Liver Automatic Segmentation (ATLAS) Dataset on Contrast-Enhanced Magnetic Resonance Imaging for Hepatocellular Carcinoma
by Félix Quinton,Romain Popoff,Benoît Presles,Sarah Leclerc,Fabrice
Meriaudeau,Guillaume Nodari,Olivier Lopez,Julie Pellegrinelli,Olivier
Chevallier,Dominique Ginhac,Jean-Marc Vrigneaud andJean-Louis Alberini
Data 2023, 8(5), 79; https://doi.org/10.3390/data8050079 - 27 Apr 2023
Viewed by 1493

Emidec: A Database Usable for the Automatic Evaluation of Myocardial Infarction from Delayed-Enhancement Cardiac MRI
by Alain Lalande,Zhihao Chen,Thomas Decourselle,Abdul Qayyum,Thibaut Pommier,Luc Lorgis,Ezequiel de la Rosa,Alexandre Cochet,Yves Cottin,Dominique Ginhac,Michel Salomon,Raphaël Couturier and Fabrice Meriaudeau
Data 2020, 5(4), 89; https://doi.org/10.3390/data5040089 - 24 Sep 2020
Cited by 32 | Viewed by 4863

Indian Diabetic Retinopathy Image Dataset (IDRiD): A Database for Diabetic Retinopathy Screening Research
by Prasanna Porwal,Samiksha Pachade,Ravi Kamble,Manesh Kokare,Girish
Deshmukh,Vivek Sahasrabuddhe and Fabrice Meriaudeau
Data 2018, 3(3), 25; https://doi.org/10.3390/data3030025 - 10 Jul 2018
Cited by 337 | Viewed by 29633

Reviewer 3 Report (Previous Reviewer 1)

1. The manuscript needs to clearly communicate the novelty and contribution of assembling the "RASTA" dataset. Reviewers noted that the introduction does not sufficiently highlight how this dataset addresses the current challenges in computer-aided diagnostic program development, especially the issue of limited open-source OCT-A acquisitions.

2. manuscript lacks an in-depth exploration of the methodology employed, particularly in regard to the Optical Coherence Tomography-Angiography (OCT-A) imaging technique. Additionally, the potential impact of the proposed approach on the field of cardiovascular disease screening and management needs to be further elucidated.

3. The manuscript should provide a comprehensive statistical analysis of the dataset and results, ensuring that the outcomes are not only significant but also robust and reproducible.

This manuscript cannot be considered for publications

Minor editing of English language required

Author Response

Reviewer 3
The manuscript needs to clearly communicate the novelty and contribution of assembling the "RASTA" dataset. Reviewers noted that the introduction does not sufficiently highlight how this dataset addresses the current challenges in computer-aided diagnostic program development, especially the issue of limited open-source OCT-A acquisitions.

We thank the Reviewer for his comments. We are not presenting an original research paper. It is an opensource dataset repository. The aim of this project is to allow collaborative work on this specific field of research. This is the first complete dataset with retinal imaging and good quality labeled cardiovascular data.

Manuscript lacks an in-depth exploration of the methodology employed, particularly in regard to the Optical Coherence Tomography-Angiography (OCT-A) imaging technique. Additionally, the potential impact of the proposed approach on the field of cardiovascular disease screening and management needs to be further elucidated.

We have already published several datasets in this journal. This dataset could motivate collaboration in this very dynamic field of research. Opensource dataset are mandatory for good quality research in Artificial Intelligence.

Retinal Fundus Multi-Disease Image Dataset (RFMiD): A Dataset for Multi-Disease Detection Research by Samiksha Pachade,Prasanna Porwal,Dhanshree Thulkar,Manesh Kokare,Girish Deshmukh,Vivek Sahasrabuddhe,Luca Giancardo,Gwenolé Quellec andFabrice Mériaudeau
Data 2021, 6(2), 14; https://doi.org/10.3390/data6020014 - 03 Feb 2021
Cited by 50 | Viewed by 14695

A Tumour and Liver Automatic Segmentation (ATLAS) Dataset on Contrast-Enhanced
Magnetic Resonance Imaging for Hepatocellular Carcinoma
by Félix Quinton,Romain Popoff,Benoît Presles,Sarah Leclerc,Fabrice
Meriaudeau,Guillaume Nodari,Olivier Lopez,Julie Pellegrinelli,Olivier
Chevallier,Dominique Ginhac,Jean-Marc Vrigneaud andJean-Louis Alberini
Data 2023, 8(5), 79; https://doi.org/10.3390/data8050079 - 27 Apr 2023
Viewed by 1493

Emidec: A Database Usable for the Automatic Evaluation of Myocardial Infarction from
Delayed-Enhancement Cardiac MRI
by Alain Lalande,Zhihao Chen,Thomas Decourselle,Abdul Qayyum,Thibaut Pommier,Luc
Lorgis,Ezequiel de la Rosa,Alexandre Cochet,Yves Cottin,Dominique Ginhac,Michel
Salomon,Raphaël Couturier and Fabrice Meriaudeau
Data 2020, 5(4), 89; https://doi.org/10.3390/data5040089 - 24 Sep 2020
Cited by 32 | Viewed by 4863

Indian Diabetic Retinopathy Image Dataset (IDRiD): A Database for Diabetic Retinopathy
Screening Research
by Prasanna Porwal,Samiksha Pachade,Ravi Kamble,Manesh Kokare,Girish
Deshmukh,Vivek Sahasrabuddhe and Fabrice Meriaudeau
Data 2018, 3(3), 25; https://doi.org/10.3390/data3030025 - 10 Jul 2018
Cited by 337 | Viewed by 29633

The manuscript should provide a comprehensive statistical analysis of the dataset and results, ensuring that the outcomes are not only significant but also robust and
reproducible.

Please refer to the submission guidelines of this specific type of manuscript in data journal.

This manuscript cannot be considered for publications 

Round 2

Reviewer 1 Report (Previous Reviewer 3)

Please, give the data in tables, Described methods are not suffucient.

Author Response

We want to thank the Reviewer for his comments and help to improve our manuscript

"Please, give the data in tables, Described methods are not suffucient."

  • The dataset could not be presented in tables. It is opensource and available for dowload at  https://rasta.u-bourgogne.fr/. We are presenting here a data descriptor
  • Regarding the described methods we are presenting here a retinal imaging data collection with cardiovascular data. The idea of our project is to share good quality images and label cardiovascular data for collaborative work and external validation. You can find in chapter 3.2 3.3 and 3.4 the retinal imaging protocol, quantitative data and cardiovascular data we used for this project. Each study protocol could be consulted upon request (registered number are accessible).

Reviewer 3 Report (Previous Reviewer 1)

Manuscript can be accepted fro publcation

Improved

Author Response

We want to thank the reviewer for your positive comment regarding the publication of this manuscript. As requested, we improved our quality of english language with a native professional medical writter.

This manuscript is a resubmission of an earlier submission. The following is a list of the peer review reports and author responses from that submission.

Round 1

Reviewer 1 Report

The research work is not novel, and the manuscript is no constructed to the journal policy. The experiments are not enough to prove the concept. I suggest rejecting this manuscript

Not good, need to improve with the help of native English speaker

Author Response

We thank the Reviewer for his comments. We submitted this manuscript as a data descriptors. Aims and Scope and authors submission guidelines for this type of manuscript were followed for Data Journal. You will find attached all information regarding data descriptors. Moreover, we decided to model previous manuscripts published as data descriptors in Data journal (https://doi.org/10.3390/data8080131, https://doi.org/10.3390/data8080128 for example)

Regarding the fact that the research work is not novel. Several datasets containing images of retinal fundus photographs have been recently publicly published (i.e. MESSIDOR, STARE project, DRIVE, E-ophtha, and EyePACS). However, SS OCT-A datasets are less widespread. To the best of our knowledge, the Retinal oct-Angiography and cardiovascular STAtus (RASTA) dataset is the first publicly available dataset that provides systematic cardiovascular data and complete SS OCT-A retinal imaging.

This work could  enhance the transparency of dataset and stimulate collaborative project with other researchers.

Here we presented a methodical papers on processes applied to data collection, treatment and analysis for future work regarding retinal imaging and cardiovascular disease.

The manuscript was fully edited and corrected by a professionnal native English medical writer Isabella Athanassiou isabella3@t-online.de

Reviewer 2 Report

The authors’ work seems very meaningful. However, the confusing structure makes it difficult to understand the author's work. Here below are some specific questions or suggestions:

1.       In the Abstract, the authors gave unnecessary details about the purpose and scope of the study. Instead, they can make a shorter and more explanatory explanation for their research, showing numerical results.

2.       Generally, the first section is an introduction. And at present, this section is too limited to show the development of this area.

3.       The author's introduction needs to be optimized, and we suggest that the author evaluate what needs to be improved in the introduction according to the following criteria.

        What is the problem to be solved?

        Are there any existing solutions?

        Which is the best?

        What is the main limitation of the best and existing approaches?

        What do you hope to change or propose to make it better?

        How is the paper structured?

4.       List of acronyms must be provided before introductions.

5.       The section of the related work should be added. In the related work, the authors must provide the design principles and basic knowledge of technical solutions. Some new and important work needs to be mentioned. These papers can provide some theoretical basis.

6.       In general, the structure of a research paper should include an introduction, related work, data/materials, methods, results, discussion, and conclusion. But the structure of this manuscript is chaotic and unclear.

7.       Most of the sections are too weak and should be revised a lot.

8.       The quality of the figures and tables need to be improved, and most of them are not adequately described and analyzed.

9.       The part of discussion needs to be added, clarifying the main academic contributions of the manuscript and comparing the differences with existing research results, without shying away from the shortcomings of the manuscript's methodology and possible paths for improvement.

10.   Please clearly highlight how your work advances the field from the present state of knowledge and you should provide a clear justification for your work. The impact or advancement of the work can also appear in the conclusion.

Moderate editing of English language required

Author Response

(The authors gave the same response as above.)

Reviewer 3 Report

Please, absolutely inadequate design. Methods at the end with no discussion? Data must be given in absolute numbers and then statsitical methods.

The necessity of grammar check form the very beginning / e.g.

the name of the Department...

poor / translated from French not adequately

Author Response

We thank the Reviewer for his comments. We submitted this manuscript as a data descriptors. Aims and Scope and authors submission guidelines for this type of manuscript were followed for Data Journal. You will find attached all information regarding data descriptors. Moreover, we decided to model previous manuscripts published as data descriptors in Data journal (https://doi.org/10.3390/data8080131, https://doi.org/10.3390/data8080128 for example)

Here we presented a methodical papers on processes applied to data collection, treatment and analysis for future work regarding retinal imaging and cardiovascular disease.

The manuscript was fully edited and corrected by a professionnal native English medical writer Isabella Athanassiou isabella3@t-online.de

We thank the Reviewer for his comment on authors' affiliations. It has been corrected in the revised version of the manuscript as follow

Department of Ophthalmology, Dijon University Hospital, 21079 Dijon CEDEX, France; clement.germanese@chu-dijon.fr (C.G.); petra.eid@chu-dijon.fr (P.E.); laure-anne.steinberg@chu-dijon.fr (LA.S.); catherine.creuzot-garcher@chu-dijon.fr (C.CG.); pierrehenry.gabrielle@chu-dijon.fr (PH.G.); louis.arnould@chu-dijon.fr (L.A.)
